# Neighborhood Makes or Breaks Active Ageing? Findings from Cross-Sectional Path Analysis

**DOI:** 10.3390/ijerph19063695

**Published:** 2022-03-20

**Authors:** Daniel R. Y. Gan, Grand H.-L. Cheng, Tze Pin Ng, Xinyi Gwee, Chang Yuan Soh, John Chye Fung, Im Sik Cho

**Affiliations:** 1Department of Gerontology, Faculty of Arts and Social Sciences, Simon Fraser University, Vancouver, BC V6B 5K3, Canada; 2Department of Medicine, Yong Loo Lin School of Medicine, National University of Singapore, Singapore 117597, Singapore; ghlcheng@nus.edu.sg; 3Department of Psychological Medicine, Yong Loo Lin School of Medicine, National University of Singapore, Singapore 117597, Singapore; pcmngtp@nus.edu.sg (T.P.N.); pcmgxy@nus.edu.sg (X.G.); 4National Kidney Foundation Singapore, Singapore 328836, Singapore; sohchangyuan@gmail.com; 5Department of Architecture, College of Design and Environment, National University of Singapore, Singapore 117566, Singapore; akifjc@nus.edu.sg (J.C.F.); akicis@nus.edu.sg (I.S.C.)

**Keywords:** social cohesion, neighborhood friendship, sense of community, mental health, pathways, older adults, neighborhood disadvantage, health-related quality of life

## Abstract

Mental ill-health prolongs and complicates other chronic illnesses, which is a major public health concern because of the potential stress it places on health systems. Prevention via active aging and place-based interventions thus became increasingly important with population aging, e.g., through health promotion and age-friendly neighborhoods. However, how the targeted outcomes of these efforts are related remains unclear. This paper examined whether the relationship between active living and mental health or health-related quality of life is mediated by neighborhood cohesion. Cross-sectional data were drawn from n = 270 community-dwelling adults aged 50 and above in the Gerontology Research Program—Center for Ageing Research in the Environment (GRP-CARE) Survey. Path analysis showed that one can live actively for better mental health (B_total_ = 0.24), but it is largely mediated by neighborhood cohesion (37%). Further examination of the factors of neighborhood cohesion showed that this mediation is explained by communal affordance (B_indirect_ = 0.05) and neighborhood friendship (B_indirect_ = 0.05). Additional study of the association between these mediators and factors of mental health revealed two psychosocial processes: (1) better community spaces (e.g., greenery and third places) support communal living (B = 0.36) and help older adults obtain emotional support (B = 0.32) for greater autonomy (B = 0.25); (2) spending more time outdoors enhances neighborhood friendship (B = 0.33) and interpersonal skills (B = 0.37), which in turn improves coping (B = 0.39). In short, the effects of active living on health are limited by one’s neighborhood environment. Neighborhood cohesion must be considered or it may stifle individual and policy efforts to age actively and healthily in urban environments. Context-sensitive implementations are required.

## 1. Introduction

Active living has long been recognized as an important avenue for older adults to age healthily, physically and psychosocially [1]. Various efforts to develop age-friendly neighborhoods aim to promote active living [2,3]. Active living may be defined as participation in various leisure, physical, and social activities, apart from work and political participation. Yet, how active living promotes the health of individuals is only partially known. Systematic reviews of the effects of the built environment on active ageing consistently point to the need for mechanistic studies to enable targeted interventions [4,5,6,7,8]. 

Extant literature on the pathways or mechanisms between neighborhood factors and the mental wellbeing of older adults highlights the role of neighborhood cohesion in addition to other neighborhood qualities [9,10,11], and health behaviors in addition to psychosocial behaviors [11,12,13,14]. Health behaviors such as walking and physical activity are often examined as mediators between neighborhood cohesion and the psychological wellbeing of older adults. Researchers also reported the mediating effect of psychosocial factors such as control [11,15], loneliness [11,16], social participation [16,17], social support [16,18], and volunteering [17].

At the same time, contextual differences must be considered to enable tailored translations based on neighborhood characteristics [19]. Important neighborhood variables that mediate its effects on wellbeing include cohesion [20,21] and safety [22]. One such study is the Jackson Heart Study, which examines of the influence of neighborhood disadvantage and cohesion on blood biomarkers [23]. The authors found that neighborhood disadvantage was correlated with neighborhood cohesion for men, and showed differential effects of active living on stress biomarkers in neighborhoods with high and low cohesion by sex. Active living was generally correlated with blood biomarkers but was not correlated for men in neighborhoods with low cohesion [23]. This suggests that the effects of active living were influenced by neighborhood cohesion. 

However, the mechanism of neighborhood cohesion on the biopsychosocial stress process remains largely unknown [24]. Barber and colleagues (2016) postulated that low neighborhood cohesion may be accompanied by crime to which men were exposed [23] (p. 113). But women in these neighborhoods with low cohesion were undeterred, which makes crime a poor explanation, as per other studies [22,25]. In light of stark socioeconomic segregation in these study settings [26], the actual explanation may be more complex. Low neighborhood cohesion and poor friendships are associated with depressive symptoms [27]. Neighborhood friendship has a positive impact on one’s wellbeing [28,29]. Various built environment attributes have been found to affect mental wellbeing through social cohesion and neighborhood quality [11]. These underscore the need to shed light on mediation pathways after controlling for neighborhood socioeconomic status [30].

To disentangle these complexities, this paper aims to examine the mediating effects of neighborhood cohesion on the relationship between active living and mental health apart from issues of safety and neighborhood disadvantage [31]. Choosing a study context that is generally safe and well-maintained allows non-crime-related pathways to be detected, which would provide clues for more targeted interventions. We ask the following question: In generally safe urban environments, what variables mediate between active living and mental and functional health? Mental and functional health were chosen as relevant goals of place-based, age-friendly interventions [3]. With reference to the Transdisciplinary Neighborhood Health Framework [32] and other studies [16,24,33], we hypothesize that neighborhood cohesion mediates between active living and mental health. We define neighborhood cohesion as the depth and breadth of social ties in a local area, regardless of societal fault lines. 

## 2. Methods

### 2.1. Participants

Cross-sectional data were collected from 270 adults aged 50 and above who were living in 270 randomly selected public housing apartments in Singapore. Inclusion criteria were adults aged 50 and above who are able to provide consent and who have lived in the neighborhood for at least three months. 

At each randomly selected apartment, a random level was selected and doors were knocked upon until an eligible and willing respondent was found. To control for neighborhood socioeconomic status, participants were selected from typical housing types. Given that most Singaporeans live in public housing and that the size of one’s dwelling unit is typically used as indicator of an individuals’ socioeconomic status, we selected participants from the median housing type of public housing apartments with predominantly three- to five-room units. These were high-rise apartments of typically 10 to 25 stories, developed and managed by the Housing and Development Board. These dwelling units are typically owner-occupied. They are safe, of good quality, and well-maintained, often with access to various amenities; high satisfaction is often reported in periodic household surveys. To minimize variations in neighborhood age structure, districts or planning areas with extreme dependency ratios were excluded. Structural experiences of aging in Singapore are discussed elsewhere [33]. 

Where possible, the survey was self-administered. Most surveys were interviewer-administered given that many respondents had received less education. Respondents were compensated a small amount for their time. These data were collected during Phase 1 of the Gerontological Research Program—Center for Ageing Research in the Environment (GRP-CARE) Survey in 2018. The study was approved by the Institutional Review Board of the National University of Singapore.

### 2.2. Measures

**Exposure variable.** Active living was measured with a five-factorial, 17-item Older People’s Active Participation scale [34]. The factors are namely socializing, physical training, listening, home-making, and outing. Respondents were asked how often they engaged in a list of activities, including aerobic exercises, body–mind exercise, card games, volunteer work, and gardening. Responses ranged from 1 (never or less than once a month) to 3 (weekly or more). It has good Cronbach’s alpha = 0.71.

**Mediating variable.** Neighborhood cohesion was measured using the four-factorial, 18-item Older People’s Neighborhood Experience (OpenX) scale, which is person-centric [35]. Factors include communal affordance, neighborhood friendship, environment pleasantness, and time outdoors. Respondents were asked to think about their everyday activity spaces near their home. Items include “There are local gathering places where residents become familiar with each other,” “The neighbors that I know also know one another,” “The spaces in my neighborhood are beautiful and inviting,” and “How often do you walk around the neighborhood for recreation.” Responses ranged from 0 (strongly disagree/not at all) to 4 (strongly agree/very often). Cronbach’s alpha = 0.83.

**Outcome variables.** Health-related quality of life (HRQOL) was measured using version 2 of the 12-item HRQOL Short Form (SF-12v2) [36]. Items include functional limitations due to physical and mental health. For example, participants were asked whether they “accomplished less than they would like at work or in their daily activities” due to physical health or emotional problems. Items on depressive symptoms and self-rated health are also included. Higher scores indicate better quality of life. Cronbach’s alpha = 0.88.

Mental wellbeing was assessed using the unidimensional six-item Rapid Positive Mental Health Instrument (R-PMHI) [37]. R-PMHI was developed through rigorous processes of context-relevant item generation, factor analyses, and psychometric assessment in a multiracial population [38]. Items include “I make friends easily” and “I am willing to share my time with others.” Responses ranged from 1 (not at all like me) to 6 (exactly like me). Higher scores indicate better mental health. Cronbach’s alpha = 0.76–0.81.

A 19-item version of the same Positive Mental Health (PMH) instrument has six factors, including autonomy, coping, emotional support, and interpersonal skills [39]. Autonomy was measured with four items (Cronbach’s alpha = 0.89), including “I am clear about what I want in life” and “I know what I need to do to reach my goals.” Coping was measured with two items (Cronbach’s alpha = 0.73), namely, “When I feel stressed, I try to solve the problem one step at a time” and “see it in a positive light”. Emotional support was measured with three items (Cronbach’s alpha = 0.80), including “When I am in a difficult situation there is someone I can rely on” and “There is someone to cheer me up if I am having a bad day”. Interpersonal skill was measured with three items (Cronbach’s alpha = 0.79), including “I get along well with others” and “I make an effort to help others”. Responses similarly ranged from 1 (not at all like me) to 6 (exactly like me). 

**Control variables.** Participants’ age, sex, and level of education were measured.

### 2.3. Statistical Analyses

Path analysis was conducted on observed variables to study mediation using Stata 14. Path analysis examined the role of neighborhood cohesion and its factors in the relationship between active living and outcome variables, controlling for age, sex, and education if correlated at zero-order [40]. Specifically, active living was adjusted for education and sex; neighborhood cohesion and emotional support were adjusted for sex; HRQOL was adjusted for age and education; time outdoors was adjusted for age; autonomy was adjusted for education. Other variables were not adjusted for as they were not correlated with the control variables. Goodness-of-fit indices were assessed. Indirect effects of active living on outcome variables were calculated as the product of the coefficients along statistically significant paths. Total effects were calculated as sum of direct and indirect effects. 

## 3. Results 

Data from 270 community-dwelling adults (64.7% female) in 270 apartment blocks were included for analysis. The average age of participants was approximately 65.4 years. Most participants completed primary (39.6%) or secondary (33.7%) education only. Detailed participant characteristics are reported elsewhere [41].

As shown Figure 1, neighborhood cohesion partially mediated the relationship between active living (B = 0.26) and mental health (B = 0.36), with an indirect effect of B_indirect_ = 0.09. There remains a direct effect (B = 0.15) between active living and mental health, yielding a total effect of B_total_ = 0.24. Neighborhood cohesion explained 0.09/0.24 = 37% of the influence of active living on mental health. Neighborhood cohesion fully mediated the relationship between active living (B = 0.26) and HRQOL (B = 0.25), resulting in a total effect of B_total_ = B_indirect_ = 0.07. Neighborhood cohesion explained 100% of the relationship between active living and HRQOL.

Further examination of the factors of neighborhood cohesion in Figure 2 showed communal affordance and neighborhood friendship mediated the relationship between active living and outcome variables. The mediating effects of environment pleasantness and time outdoors were not found. Full mediation was found; there was no direct effect (B_direct_) of active living on the outcome variables. Communal affordance mediated the relationship between active living (B = 0.19) and mental health (B = 0.24) and HRQOL (B = 0.16), yielding indirect effects of B_indirect_ = 0.05 and 0.03, respectively. Neighborhood friendship mediated only the relationship between active living (B = 0.23) and mental health (B = 0.22), yielding indirect effect of the same magnitude B_indirect_ = 0.05. 

Combining findings from both models, communal affordance and neighborhood friendship each explained 0.05/0.24 = 21% of the effects on mental health. Communal affordance explained 0.03/0.07 = 43% of the effects on HRQOL. 

A third model was constructed to further examine the relationship between communal affordance and neighborhood friendship and factors of mental health. Communal affordance was associated with emotional support (B = 0.32), and together they mediated the relationship between environment pleasantness (B = 0.36) and autonomy (B = 0.25), yielding indirect effects of B_indirect_ = 0.03 (16%). Neighborhood friendship was associated with interpersonal skills (B = 0.37), and together they mediated the relationship from environment pleasantness (B = 0.28) and time outdoors (B = 0.33) to autonomy (B = 0.29) and coping (B = 0.39), yielding indirect effects of B_indirect_ = 0.03 + 0.04 + 0.04 + 0.05 = 0.16 (84%). No other paths were found (See Figure 3).

Acceptable goodness-of-fit indices were found. In both models, comparative fit index (CFI) and Tucker–Lewis index (TLI) were 0.97 and 0.94, respectively, above the acceptable value of 0.90; root mean square error of approximation (RMSEA) was 0.04 with an upper bound of 0.08, whereas standardized root mean square residual (SRMR) was 0.04, both below the acceptable threshold of 0.10; chi-square was also not significant, as would be required for good model fit [42]. Similar model fit was found for the third model (TLI = 0.996, CFI = 0.994, RMSEA = 0.01, SRMR = 0.05, chi-square *p*-value = 0.406).

## 4. Discussion

This study examined the role of neighborhood cohesion as a mediator between healthy behaviors and health outcomes among adults aged 50 and above in safe urban environments, apart from the influences of neighborhood disorder. Active ageing was promoted as a more holistic framework for health promotion among older adults beyond traditional health behaviors [1,2,3]. While active living was operationalized as leisure-time physical activity in some studies [23], we used a measure that includes a variety of activities in the everyday lives of older adults in addition to physical training. Instances of full mediation coupled with good model fit suggest that neighborhood cohesion is indeed an important mechanism of place-based, age-friendly efforts [16,21,32]. Urban older adults who engage in various activities frequently are much more likely to experience better mental health and wellbeing when these activities are accompanied by a cohesive environment, with 37–100% mediation. Specifically, communal affordance and neighborhood friendship are relevant targets for interventions. That said, neighborhood safety must first be addressed [22,25]. 

Our findings reinforce the need for an environmental wellness model of health [43] especially when addressing the health of older adults at a population level. Not all neighborhood environments are supportive of older adults’ health. Active aging needs to be complemented with communal provisions to promote health. Policymakers must assess the neighborhood environment while promoting active aging [44], including the perceptions of older adults [8]. The mediating effect of communal affordances (opportunities) requires policymakers to examine whether sufficient social infrastructure and/or civic organizations are available in each neighborhood to promote neighborhood friendship. Existing conditions in various neighborhoods are to be assessed and improved before deploying aging-in-place as a policy solution. Interventions such as Neighbor Day [45] or telephone outreach [46] can be necessary in areas where neighborhood cohesion is low [23]. Stepped-wedge controlled trial may be useful to guide evidence-based improvements to interventions [47].

The effect of active living on mental health was fully explained by communal affordance and neighborhood friendship. In other words, the only reason active living enhances mental health is because active living enables communal living and increases belonging near one’s home. In light of this finding, it is unclear why the direct effect from active living to mental health had remained in the first model. It is possible that the practice of living actively inherently constitutes a part of mental health, independent of one’s experiences of the neighborhood. Other factors such the quality of home environment and family relationships may also influence the effectiveness of active living to promote mental health [48]. Future research could examine neurobiological [49,50], psychosocial [51,52], and ecological processes [52,53].

Further analysis of the paths between factors of neighborhood cohesion and mental health revealed two distinct neighborhood health processes: (1) communal affordance → emotional support, and (2) neighborhood friendship → interpersonal skills. Better community spaces, such as greenery and third places, support communal living and help older adults obtain emotional support. Spending more time outdoors enhances neighborhood friendship and interpersonal skills, which in turn improves coping. The neighborhood environment has been shown to provide opportunities to form interpersonal relationships that may be beneficial for older adults’ mental wellbeing [41,54,55]. Practicing one’s interpersonal skills may enable older adults to remain engaged, adapt to negative social situations, and grow as individuals. 

These pathways reveal points of intervention that may support the mental wellbeing of older adults living in community. To aid the process of “creating connections, empowerment, social influence, and access to material resources and services” [56] (p.103) in the community, it is vital to uncover the neighborhood health processes that empower older adults with greater autonomy. Social support has been found to explain the relationship between neighborhood problems and negative emotions [18]. Emotional assurances of one’s worth likely increase one’s ability to cope amid late-life stresses and to retain one’s sense of agency. Community spaces may be created in neighborhoods where such amenities are missing [57].

### 4.1. Implications

On the policy level, the above findings limit the applicability of aging-in-place as a policy solution. In the face of rapid aging, health economists and policymakers are rightly concerned with the potential surges in the cost of healthcare and required facilities, whether with or without financing mechanisms. Recent efforts to “reframe aging” challenge these alarmist responses to demographic changes as they may have unintended effects on the wellbeing of older adults [58]. From this perspective, aging-in-place attracted policy interest because it greatly reduces the cost required to provide adequate facilities. Yet, insofar as some neighborhoods remain socially fractured and do not provide opportunities for meaningful communal living, aging in these places will likely be counterproductive.

Mental health benefits from the neighborhood may remain accessible only to those who already had well-established neighborhood friendship before the COVID-19 pandemic [29,59]. Community-dwelling older adults in isolation may require innovative ways to compensate for the losses of communal living that once provided emotional support [60,61].

### 4.2. Limitations

The study used cross-sectional data. Although cross-sectional data may support causal inferences given acceptance of a set of theoretic arguments used to construct a structural equation model [62,63], the use of longitudinal and experimental data will better dispel doubts on causality. Cross-lagged analysis may examine bidirectional causality. The study was conducted at high-density public housing neighborhoods in Singapore, which are generally safe, well-maintained, desegregated, and free from environmental pollution. Pilot interventions should be conducted before scaling up in other urban areas with similar conditions. The applicability of this study’s findings in less advantaged neighborhoods with different characteristics remains to be tested.

## 5. Conclusions

This study showed that active living enhances mental health and wellbeing among older adults in urban neighborhoods through neighborhood cohesion, specifically communal affordance and neighborhood friendship. Good neighborhood environments provide an avenue to receive emotional support and hone interpersonal skills. These findings refine current approaches to promote aging-in-place, and call for place-based interventions to complement active aging campaigns so that these health promotion efforts can be effective in otherwise unsupportive neighborhood environments [64].

Aging has often been discussed in relation to macrolevel, structural factors such as healthcare financing, or microlevel factors such as care of individuals with dementia. The missing middle “meso-level” scale of the neighborhood is important as a site of exposure and intervention with reference to the Transdisciplinary Neighborhood Health Framework [32]. Context-sensitive and careful implementations are required based on psychosocial insights. Neighborhood cohesion must be considered or it may stifle individual and policy efforts to age actively and healthily.

## Figures and Tables

**Figure 1 ijerph-19-03695-f001:**
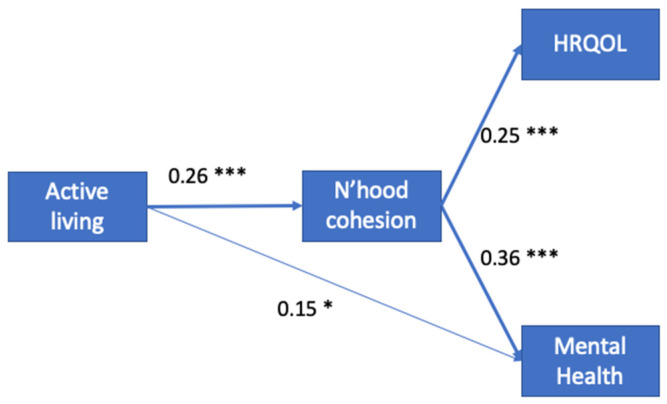
Mediation analysis through neighborhood cohesion. Path coefficients are shown. (n = 270; *** *p* < 0.001; ** *p* < 0.01; * *p* < 0.05).

**Figure 2 ijerph-19-03695-f002:**
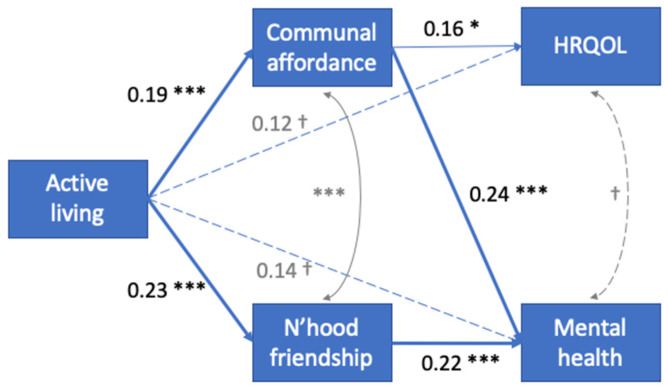
Mediation analysis through neighborhood cohesion. Path coefficients are shown (n = 270; *** *p* < 0.001; ** *p* < 0.01; * *p* < 0.05; † *p* < 0.10).

**Figure 3 ijerph-19-03695-f003:**
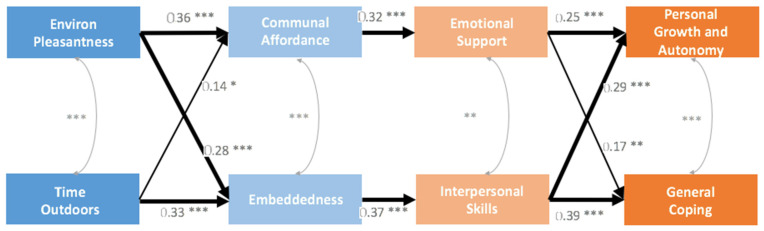
Neighborhood health processes, as shown in path analysis from factors of neighborhood cohesion to factors of positive mental health. Path coefficients are shown (n = 202; *** *p* < 0.001; ** *p* < 0.01; * *p* < 0.05). Embeddedness = Neighborhood Friendship.

## Data Availability

Data and study material are available from the GRP-CARE Survey website: https://blog.nus.edu.sg/healthyageinginplace/data/ (accessed on 30 June 2018).

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
