# Peer review of "Neighborhood Makes or Breaks Active Ageing? Findings from Cross-Sectional Path Analysis"

_ijerph, 2022, doi:10.3390/ijerph19063695_

Round 1

Reviewer 1 Report

Title:

Add 'in Singapore' to the title

Abstract

It is not a normal practice to provide citation in the abstract. 

Need to add more information in the abstract on the aims and methods of the study.

Add summary information on the place of study and participants

Introduction

Add definitions of active living and community cohesion.

Please use the introduction to provide some context specific about Singapore and its ageing population.

Method

sec 2.1 please add a table with participant characteristics.

Adequate selection of SEM to examine mediation effects. Please specify the exact outcome variables that were examined simultaneously in the model in this section.

Results

The results shows a model with HRQOL and mental health as the outcome variables but the method subsection 2.3 implies that the two variables are HRQOL and active ageing. Please clarify in section 2.3

Discussion

Please reflect on the findings within the context of Singapore and family/community living arrangements for example

Reviewer 2 Report

Thank you for an interesting paper. I have the following comments.

Abstract: Please exclude the reference provided. I suggest the abstract is revised so that it comprises more of Methods, Results, Conclusion and less text related to the Introduction. 

Introduction: The authors might want to address previous research related to mental health and activity. This would add to the quality of the paper, however, it is not necessary to do so. 

Methods:

The first paragraph, Participants, would benefit from more structure. It includes also Data collection procedures which preferrably could be moved to another paragraph. 

Exposure variables: Measurement of active living requires further explanation. Which instrument was used? Description of psychometric properties required.

Mediating and outcome variables: Please provide a description of the psychometric properties of each instrument, if not included yet. 

Conclusion: The sentence comprising reference to the Transdisciplinary Neighborhood Health Framework is redundant and would better fit in the Introduction section OR as a framework for analysing and organising the presentation of the data. Another suggestion is to remove it since it does not particularly add to the paper if not used throughout. 

Reviewer 3 Report

Major point

The authors continue to say that active ageing improves well-being ONLY IF cohesion is high. This, however, is not what your data demonstrate: they show a mediation (A causes B through C), while the "ONLY IF" is a moderation hypothesis (A causes B only when C is present), which they do not test.

Minor point

The Authors test a so-called multilevel mediation model since the predictor (active ageing) is at the individual level, cohesion is at the neighbourhood/collective level, and the outcome (well-being) is again at the individual level. However, they do not adopt methodologies to take this into account, and therefore the results may be distorted. However, it seems to me that there are no reliable statistical models to calculate the effect of a predictor at the low level (individual) on a high-level outcome (the neighbourhood), so I don’t even know how they could do, even if I wanted to.  

Round 2

Reviewer 2 Report

Accept for publication

Reviewer 3 Report

The authors have correctly reset the paper, modifying the statistics.